# Provable Training for Graph Contrastive Learning

**Yue Yu[1], Xiao Wang[2]\*, Mengmei Zhang[1], Nian Liu[1], Chuan Shi[1]\***
[1]Beijing University of Posts and Telecommunications, China
[2] Beihang University, China
`yuyue1218@bupt.edu.cn, xiao_wang@buaa.edu.cn,`
`{zhangmm, nianliu, shichuan}@bupt.edu.cn`

## Abstract

Graph Contrastive Learning (GCL) has emerged as a popular training approach for learning node embeddings from augmented graphs without labels. Despite the key principle that maximizing the similarity between positive node pairs while minimizing it between negative node pairs is well established, some fundamental problems are still unclear. *Considering the complex graph structure, are some nodes consistently well-trained and following this principle even with different graph augmentations? Or are there some nodes more likely to be untrained across graph augmentations and violate the principle? How to distinguish these nodes and further guide the training of GCL?* To answer these questions, we first present experimental evidence showing that the training of GCL is indeed imbalanced across all nodes. To address this problem, we propose the metric "node compactness", which is the lower bound of how a node follows the GCL principle related to the range of augmentations. We further derive the form of node compactness theoretically through bound propagation, which can be integrated into binary cross-entropy as a regularization. To this end, we propose the PrOvable Training (POT[1]) for GCL, which regularizes the training of GCL to encode node embeddings that follows the GCL principle better. Through extensive experiments on various benchmarks, POT consistently improves the existing GCL approaches, serving as a friendly plugin.

## 1 Introduction

Graph Neural Networks (GNNs) are successful in many graph-related applications, e.g., node classification [10, 20] and link prediction [33, 27]. Traditional GNNs follow a semi-supervised learning paradigm requiring task-specific high-quality labels, which are expensive to obtain [9]. To alleviate the dependence on labels, graph contrastive learning (GCL) is proposed and has received considerable attention recently [35, 31, 25, 21, 29]. As a typical graph self-supervised learning technique, GCL shows promising results on many downstream tasks [22, 30, 23].

The learning process of GCL usually consists of the following steps: generate two graph augmentations [35]; feed these augmented graphs into a graph encoder to learn the node embeddings, and then optimize the whole training process based on the InfoNCE principle [15, 35, 36], where the positive and negative node pairs are selected from the two augmentations. With this principle ("GCL principle" for short), GCL learns the node embeddings where the similarities between the positive node pairs are maximized while the similarities between the negative node pairs are minimized.

However, it is well known that the real-world graph structure is complex and different nodes may have different properties [1, 2, 11, 36]. When generating augmentations and training on the complex graph

---
\*Corresponding authors.
[1]Code available at https://github.com/VoidHaruhi/POT-GCL

structure in GCL, it may be hard for all nodes to follow the GCL principle well enough. This naturally gives rise to the following questions: *are some of the nodes always well-trained and following the principle of GCL given different graph augmentations? Or are there some nodes more likely not to be well trained and violate the principle?* If so, it implies that each node should not be treated equally in the training process. Specifically, GCL should pay less attention to those well-trained nodes that already satisfy the InfoNCE principle. Instead, GCL should focus more on those nodes that are sensitive to the graph augmentations and hard to be well trained. The answer to these questions can not only deepen our understanding of the learning mechanism but also improve the current GCL. We start with an experimental analysis (Section 3) and find that the training of GCL is severely imbalanced, i.e. the averaged InfoNCE loss values across the nodes have a fairly high variance, especially for the not well-trained nodes, indicating that not all the nodes follow the GCL principle well enough.

Once the weakness of GCL is identified, another two questions naturally arise: *how to distinguish these nodes given so many possible graph augmentations? How to utilize these findings to further improve the existing GCL?* Usually, given a specific graph augmentation, it is easy to check how different nodes are trained to be. However, the result may change with respect to different graph augmentations, since the graph augmentations are proposed from different views and the graph structure can be changed in a very diverse manner. Therefore, it is technically challenging to discover the sensitivity of nodes to all these possible augmentations.

In this paper, we theoretically analyze the property of different nodes in GCL. We propose a novel concept "node compactness", measuring how worse different nodes follow the GCL principle in all possible graph augmentations. We use a bound propagation process [5, 32] to theoretically derive the node compactness, which depends on node embeddings and network parameters during training. We finally propose a novel PrOvable Training model for GCL (POT) to improve the training of GCL, which utilizes node compactness as a regularization term. The proposed POT encourages the nodes to follow the GCL principle better. Moreover, because our provable training model is not specifically designed for some graph augmentation, it can be used as a friendly plug-in to improve current different GCL methods. To conclude, our contributions are summarized as follows:

- We theoretically analyze the properties of different nodes in GCL given different graph augmentations, and discover that the training of GCL methods is severely imbalanced, i.e., not all the nodes follow the principle of GCL well enough and many nodes are always not well trained in GCL.

- We propose the concept of "node compactness" that measures how each node follows the GCL principle. We derive the node compactness as a regularization using the bound propagation method and propose PrOvable Training for GCL (POT), which improves the training process of GCL provably.

- POT is a general plug-in and can be easily combined with existing GCL methods. We evaluate our POT method on various benchmarks, well showing that POT consistently improves the current GCL baselines.

## 2 Preliminaries

Let $G = (\mathcal{V}, \mathcal{E})$ denote a graph, where $\mathcal{V}$ is the set of nodes and $\mathcal{E} \subseteq \mathcal{V} \times \mathcal{V}$ is the set of edges, respectively. $\mathbf{X} \in \mathbb{R}^{N \times F}$ and $\mathbf{A} = \{0, 1\}^{N \times N}$ are the feature matrix and the adjacency matrix. $\mathbf{D} = diag\{d_1, d_2, \ldots, d_N\}$ is the degree matrix, where each element $d_i = \|\mathbf{A}_i\|_0 = \sum_{j \in \mathcal{V}} a_{ij}$ denotes the degree of node $i$, $\mathbf{A}_i$ is the $i$-th row of $\mathbf{A}$. Additionally, $\mathbf{A}_{sym} = \mathbf{D}^{-1/2}(\mathbf{A} + \mathbf{I})\mathbf{D}^{-1/2}$ is the degree normalized adjacency matrix with self-loop (also called the message-passing matrix).

GCN [10] is one of the most common encoders for graph data. A graph convolution layer can be described as $\mathbf{H}^{(k)} = \sigma(\mathbf{A}_{sym}\mathbf{H}^{(k-1)}\mathbf{W}^{(k)})$, where $\mathbf{H}^{(k)}$ and $\mathbf{W}^{(k)}$ are the embedding and the weight matrix of the $k$-th layer respectively, $\mathbf{H}^{(0)} = \mathbf{X}$, and $\sigma$ represents the non-linear activation function. A GCN encoder $f_\theta$ with $K$ layers takes a graph $(\mathbf{X}, \mathbf{A})$ as input and returns the embedding $\mathbf{Z} = f_\theta(\mathbf{X}, \mathbf{A})$.

**Graph Contrastive learning (GCL)**. GCL learns the embeddings of the nodes in a self-supervised manner. We consider a popular setting as follows. First, two graph augmentations $G_1 = (\mathbf{X}, \tilde{\mathbf{A}}_1)$ and $G_2 = (\mathbf{X}, \tilde{\mathbf{A}}_2)$ are sampled from $\mathcal{G}$, which is the set of all possible augmentations. In this paper,

we focus on the augmentations to graph topology (e.g. edge dropping [35]). Then the embeddings $\mathbf{Z}_1 = f_\theta(\mathbf{X}, \tilde{\mathbf{A}}_1)$ and $\mathbf{Z}_2 = f_\theta(\mathbf{X}, \tilde{\mathbf{A}}_2)$ are learned with a two-layer GCN [35, 36]. InfoNCE [15, 35] is the training objective as follows:

$$l(\mathbf{z}_{1,i},\ \mathbf{z}_{2,i}) = \frac{e^{\theta(\mathbf{z}_{1,i},\mathbf{z}_{2,i})/\tau}}{e^{\theta(\mathbf{z}_{1,i},\mathbf{z}_{2,i})/\tau} + \sum_{j \neq i} e^{\theta(\mathbf{z}_{1,i},\mathbf{z}_{2,j})/\tau} + \sum_{l \neq i} e^{\theta(\mathbf{z}_{1,i},\mathbf{z}_{1,l})/\tau}}, \tag{1}$$

where $\theta(a,b) = s(g(a),\ g(b))$, $s$ is a similarity measure, $g$ is a projector, and $\tau$ is the temperature hyperparameter. For node $i$, the positive pair is $(\mathbf{z}_{1,i},\ \mathbf{z}_{2,i})$, where $\mathbf{z}_{1,i}$ and $\mathbf{z}_{2,i}$ denote the embedding of node $i$ in the two augmentations, $\{(\mathbf{z}_{1,i}, \mathbf{z}_{2,j}) \mid j \neq i\}$ and $\{(\mathbf{z}_{1,i}, \mathbf{z}_{1,l}) \mid l \neq i\}$ are the negative pairs. The overall loss function of the two augmentations is $\mathcal{L}_{\text{InfoNCE}}(\mathbf{Z}_1, \mathbf{Z}_2) = \frac{1}{2N} \sum_{i=1}^{N} (l(\mathbf{z}_{1,i}, \mathbf{z}_{2,i}) + l(\mathbf{z}_{2,i}, \mathbf{z}_{1,i}))$.

# 3 The Imbalanced Training of GCL: an Experimental Study

As mentioned in Section 1, due to the complex graph structure, we aim to investigate whether all the nodes are trained to follow the principle of GCL. Specifically, we design an experiment by plotting the distribution of InfoNCE loss values of the nodes. To illustrate the result of a GCL method, we sample 500 pairs of augmented graphs with the same augmentation strategy in the GCL method and obtain the average values of In-

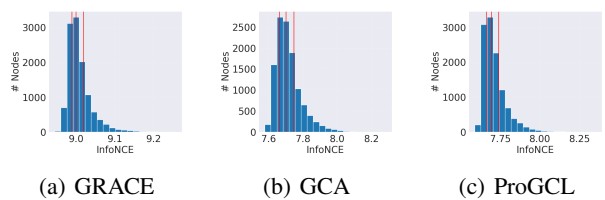

(a) GRACE      (b) GCA      (c) ProGCL

Figure 1: The imbalance of GCL training

foNCE loss. We show the results of GRACE [35], GCA [36], and ProGCL [25] on WikiCS [14] in Figure 1 with histograms. The vertical red lines represent 1/4-quantile, median, and 3/4-quantile from left to right. The results imply that the nodes do not follow the GCL principle well enough in two aspects: 1) the InfoNCE loss of the nodes has a high variance; 2) It is shown that the untrained nodes (with higher InfoNCE loss values) are much further away from the median than the well-trained nodes (with lower InfoNCE loss values), indicating an imbalanced training in the untrained nodes. GCL methods should focus more on those untrained nodes. Therefore, it is highly desired that we should closely examine the property of different nodes so that the nodes can be trained as we expected in a provable way. Results on more datasets can be found in Appendix D.

# 4 Methodology

## 4.1 Evaluating How the Nodes Follow the GCL Principle

In this section, we aim to define a metric that measures the extent of a node following the GCL principle. Since how the nodes follow the GCL principle relies on the complex graph structure generated by augmentation, the metric should be related to the set of all possible augmentations. Because InfoNCE loss can only measure the training of a node under two pre-determined augmentations, which does not meet the requirement, we propose a new metric as follows:

**Definition 1** (Node Compactness). *Let $f$ denote the encoder of Graph Contrastive Learning. The compactness of node $i$ is defined as*

$$\underline{\tilde{f}}_{1,i} = \min_{\substack{G_1=(\mathbf{X},\mathbf{A}_1)\in\mathcal{G};\\ G_2=(\mathbf{X},\mathbf{A}_2)\in\mathcal{G}}} \left\{ \frac{1}{N-1} \sum_{j\neq i} (\mathbf{z}_{1,i}\cdot\bar{\mathbf{z}}_{2,i} - \mathbf{z}_{1,i}\cdot\bar{\mathbf{z}}_{2,j}) \mid \mathbf{Z}_1 = f_\theta(\mathbf{X},\mathbf{A}_1), \mathbf{Z}_2 = f_\theta(\mathbf{X},\mathbf{A}_2) \right\}. \tag{2}$$

*where $\bar{\mathbf{z}} = \mathbf{z}/\left\|\mathbf{z}\right\|_2$ is the normalized vector.*

Definition 1 measures how worse node $i$ follows the GCL principle in all possible augmentations. If $\underline{\tilde{f}}_i$ is greater than zero, on average, the node embedding $\mathbf{z}_{1,i}$ is closer to the embedding of the positive sample $\mathbf{z}_{2,i}$ than the embedding of the negative sample $\mathbf{z}_{2,j}$ in the worst case of augmentation; otherwise, the embeddings of negative samples are closer. Note that normalization is used to eliminate

the effect of a vector's scale. In Definition 1, the anchor node is node $i$ in $G_1$, so similarly we can define $\underline{\tilde{f}}_{2,i}$ if the anchor node is node $i$ in $G_2$.

We can just take one of $G_1$ and $G_2$ as the variable, for example, set $G_2$ to be a specific sampled augmentation and $G_1$ to be an augmentation within the set $\mathcal{G}$. From this perspective, based on Definition 1, we can also obtain a metric, which has only one variable:

**Definition 2** ($G_2$-Node Compactness). *Under Definition 1, if $G_2$ is the augmentation sampled in an epoch of GCL training, then the $G_2$-Node Compactness is*

$$\underline{\tilde{f}}_{G_2,i} = \min_{G_1 = f_\theta(\mathbf{X}, \mathbf{A}_1) \in \mathcal{G}} \{\mathbf{z}_{1,i} \cdot \mathbf{W}_{2,i}^{CL} \mid \mathbf{Z}_1 = f_\theta(\mathbf{X}, \mathbf{A}_1)\}, \tag{3}$$

*where $\mathbf{W}_{2,i}^{CL} = \bar{\mathbf{z}}_{2,i} - \frac{1}{N-1}\sum_{j \neq i} \bar{\mathbf{z}}_{2,j}$ is a constant vector.*

Similarly, we can define $G_1$-node compactness $\underline{\tilde{f}}_{G_1,i}$. Usually we have $\underline{\tilde{f}}_{G_1,i} \geq \underline{\tilde{f}}_{2,i}$ and $\underline{\tilde{f}}_{G_2,i} \geq \underline{\tilde{f}}_{1,i}$, i.e., $G_1$-node compactness and $G_2$-node compactness are upper bounds of node compactness.

## 4.2 Provable Training of GCL

To enforce the nodes to better follow the GCL principle, we optimize $G_1$-node compactness and $G_2$-node compactness in Definition 2. Designing the objective can be viewed as implicitly performing a "binary classification": the "prediction" is $\min_{G_1} z_{1,i} \cdot W_{2,i}^{CL}$, and the "label" is $\mathbf{1}$, which means that we expect the embedding of the anchor more likely to be classified as a positive sample in the worst case. To conclude, we propose the PrOvable Training (POT) for GCL, which provably enforces the nodes to be trained more aligned with the GCL principle. Specifically, we use binary cross-entropy as a regularization term:

$$\mathcal{L}_{\text{POT}}(G_1, G_2) = \frac{1}{2}(\mathcal{L}_{\text{BCE}}(\underline{\tilde{f}}_{G_1}, \mathbf{1}_n) + \mathcal{L}_{\text{BCE}}(\underline{\tilde{f}}_{G_2}, \mathbf{1}_n)), \tag{4}$$

where $\mathcal{L}_{\text{BCE}}(x, y) = -\sum_{i=1}^{N}(y_i \log \sigma(x_i) + (1 - y_i) \log(1 - \sigma(x_i)))$, $\sigma$ is the sigmoid activation, $\mathbf{1}_N = [1, 1, \ldots, 1] \in \mathbb{R}^N$, and $\underline{\tilde{f}}_{G_2} = [\underline{\tilde{f}}_{G_2,1}, \underline{\tilde{f}}_{G_2,2}, \ldots, \underline{\tilde{f}}_{G_2,N}]$ is the vector of the lower bounds in Eq. 3. Overall, we add this term to the original InfoNCE objective and the loss function is

$$\mathcal{L} = (1 - \kappa)\mathcal{L}_{\text{InfoNCE}}(Z_1, Z_2) + \kappa \mathcal{L}_{\text{POT}}(G_1, G_2), \tag{5}$$

where $\kappa$ is the hyperparameter balancing the InfoNCE loss and the compactness loss. Since $G_1$-node compactness and $G_2$-node compactness are related to the network parameter $\theta$, $\mathcal{L}_{\text{POT}}$ can be naturally integrated into the loss function, which can regularize the network parameters to encode node embeddings more likely to follow the GCL principle better.

## 4.3 Deriving the Lower Bounds

In Section 4.2, we have proposed the formation of our loss function, but the lower bounds $\underline{\tilde{f}}_{G_1}$ and $\underline{\tilde{f}}_{G_2}$ are still to be solved. In fact, to see Eq. 3 from another perspective, $\underline{\tilde{f}}_{G_1,1}$ (or $\underline{\tilde{f}}_{G_2,1}$) is the lower bound of the output of the neural network concatenating a GCN encoder and a linear projection with parameter $\mathbf{W}_{1,i}^{CL}$ (or $\mathbf{W}_{2,i}^{CL}$). It is difficult to find the particular $G_1$ which minimizes Eq. 3, since finding that exact solution $G_1$ can be an NP-hard problem considering the discreteness of graph augmentations. Therefore, inspired by the bound propagation methods [5, 32, 38], we derive the output lower bounds of that concatenated network in a bound propagation manner, i.e., we can derive the lower bounds $\underline{\tilde{f}}_{G_1}$ and $\underline{\tilde{f}}_{G_2}$ by propagating the bounds of the augmented adjacency matrix $\mathbf{A}_1$.

However, there are still two challenges when applying bound propagation to our problem: 1) $\mathbf{A}_1$ in Eq. 3 is not defined as continuous values with element-wise bounds; 2) to do the propagation process, the nonlinearities in the GCN encoder $f_\theta$ should be relaxed.

**Defining the adjacency matrix with continuous values** First, we define the set of augmented adjacency matrices as follows:

**Definition 3** (Augmented adjacency matrix). *The set of augmented adjacency matrices is*

$$\mathcal{A} = \{\tilde{\mathbf{A}} \in \{0,1\}^{N \times N} | \tilde{a}_{ij} \leq a_{ij} \wedge \tilde{\mathbf{A}} = \tilde{\mathbf{A}}^T$$
$$\wedge \|\tilde{\mathbf{A}} - \mathbf{A}\|_0 \leq 2Q \wedge \left\|\tilde{\mathbf{A}}_i - \mathbf{A}_i\right\|_0 \leq q_i \ \forall 1 \leq i \leq N\},$$

*where $\tilde{\mathbf{A}}$ denotes the augmented adjacency matrix, $Q$ and $q_i$ are the global and local budgets of edge dropped, which are determined by the specific GCL model.*

The first constraint restricts the augmentation to only edge dropping, which is a common setting in existing GCL models[35, 36, 8, 25](see more discussions in Appendix B). The second constraint is the symmetric constraint of topology augmentations.

Since the message-passing matrix is used in GNN instead of the original adjacency matrix, we further relax Definition 3 and define the set of all possible message-passing matrices, which are matrices with continuous entries as expected, inspired by [38]:

**Definition 4** (Augmented message-passing matrix). *The set of augmented message-passing matrices is*

$$\hat{\mathcal{A}} = \{\hat{\mathbf{A}} \in [0,1]^{N \times N} \mid \hat{\mathbf{A}} = \hat{\mathbf{A}}^T \wedge \forall i,j : L_{ij} \leq \hat{a}_{ij} \leq U_{ij}\},$$
$$L_{ij} = \left\{ \begin{array}{ll} \tilde{a}_{ij} & ,i = j \\ 0 & ,i \neq j \end{array} \right., \ U_{ij} = \left\{ \begin{array}{ll} ((d_i - q_i)(d_j - q_j))^{-\frac{1}{2}} & ,i = j \\ \min\{a_{ij}, ((d_i - q_i)(d_j - q_j))^{-\frac{1}{2}}\} & ,i \neq j \end{array} \right.,$$

*where $\hat{\mathbf{A}}$ denotes the augmented message-passing matrix.*

Under Definition 4, the augmented message-passing matrices are non-discrete matrices with element-wise bounds $L_{ij}$ and $U_{ij}$. $L_{ij}$ is $\tilde{a}_{ij}$ when $i = j$, since there is no edge added and the self-loop can not be dropped; $L_{ij}$ is 0 when $i \neq j$, which implies the potential dropping of each edge. $U_{ij}$ is set to $((d_i - q_i)(d_j - q_j))^{-\frac{1}{2}}$ when $i = j$ because $\hat{a}_{ii}$ is the largest when its adjacent edges are dropped as much as possible, and $U_{ij} = \min\{a_{ij}, ((d_i - q_i)(d_j - q_j))^{-\frac{1}{2}}\}$ when $i \neq j$ since there is only edge dropping, $U_{ij}$ cannot exceed $a_{ij}$.

**Relaxing the nonlinearities in the GCN encoder**    The GCN encoder in our problem has non-linear activation functions as the nonlinearities in the network, which is not suitable for bound propagation. Therefore, we relax the activations with linear bounds [32]:

**Definition 5** (Linear bounds of non-linear activation function). *For node $i$ and the $m$-th neuron in the $k$-th layer, the input of the neuron is $p_{i,m}^{(k)}$, and $\sigma$ is the activation function of the neuron. With the lower bound and upper bound of the input denoted as $p_{L,i,m}^{(k)}$ and $p_{U,i,m}^{(k)}$, the output of the neuron is bounded by linear functions with parameters $\alpha_{L,i,m}^{(k)}, \alpha_{U,i,m}^{(k)}, \beta_{L,i,m}^{(k)}, \beta_{U,i,m}^{(k)} \in \mathbb{R}$, i.e.*

$$\alpha_{L,i,m}^{(k)}(p_{i,m}^{(k)} + \beta_{L,i,m}^{(k)}) \leq \sigma(p_{i,m}^{(k)}) \leq \alpha_{U,i,m}^{(k)}(p_{i,m}^{(k)} + \beta_{U,i,m}^{(k)}).$$

*The values of $\alpha_{L,i,m}^{(k)}, \alpha_{U,i,m}^{(k)}, \beta_{L,i,m}^{(k)}, \beta_{U,i,m}^{(k)}$ depend on the activation $\sigma$ and the pre-activation bounds $p_{L,i,m}^{(k)}$ and $p_{U,i,m}^{(k)}$.*

To further obtain $\alpha$ and $\beta$, we compute the pre-activation bounds $\mathbf{P}_L^{(k)}$ and $\mathbf{P}_U^{(k)}$ as follows, which is the matrix with elements $p_{L,i,m}^{(k)}$ and $p_{U,i,m}^{(k)}$:

**Theorem 1** (Pre-activation bounds of each layer [16]). *If $f = \sigma(\hat{\mathbf{A}}\sigma(\hat{\mathbf{A}}\mathbf{X}\mathbf{W}^{(1)} + \mathbf{b}^{(1)})\mathbf{W}^{(2)} + \mathbf{b}^{(2)})$ is the two-layer GCN encoder, given the element-wise bounds of $\hat{\mathbf{A}}$ in Definition 4, $\mathbf{H}^{(k)}$ is the input embedding of $t$-th layer, then the pre-activation bounds of $t$-th layer are*

$$\mathbf{P}_L^{(k)} = \mathbf{L}[\mathbf{H}^{(k)}\mathbf{W}^{(k)}]_+ + \mathbf{U}[\mathbf{H}^{(k)}\mathbf{W}^{(k)}]_- + \mathbf{b}^{(k)},$$
$$\mathbf{P}_U^{(k)} = \mathbf{U}[\mathbf{H}^{(k)}\mathbf{W}^{(k)}]_+ + \mathbf{L}[\mathbf{H}^{(k)}\mathbf{W}^{(k)}]_- + \mathbf{b}^{(k)},$$

*where $[\mathbf{X}]_+ = \max(\mathbf{X}, 0)$ and $[\mathbf{X}]_- = \min(\mathbf{X}, 0)$.*

The detailed proof is provided in Appendix A.1. Therefore, $p_{L,i,m}^{(k)}$ and $p_{U,i,m}^{(k)}$ are known with Theorem 1 given, then the values of $\alpha$ and $\beta$ can be obtained by referring to Table 1. In the table, we denote $p_{U,i,m}^{(k)}$ as $u$ and $p_{L,i,m}^{(k)}$ as $l$ for short. $\gamma$ is the slope parameter of ReLU-like activation functions that are usually used in GCL, such as ReLU, PReLU, and RReLU. For example, $\gamma$ is zero for ReLU.

To this end, all the parameters to derive the lower bounds $\underline{\tilde{f}}_{G_1}$ and $\underline{\tilde{f}}_{G_2}$ are known: the bounds of the variable $\hat{\mathbf{A}}$ in our problem, which are $\mathbf{L}$ and $\mathbf{U}$ defined in Definition 4; the parameters of the linear bounds for non-linear activations, which are $\alpha$ and $\beta$ defined in Definition 5 and obtained by Theorem 1 and Table 1. To conclude, the whole process is summarized in the following theorem, where we apply bound propagation to our problem with those parameters:

Table 1: Values of $\alpha$ and $\beta$

|  | $l \geq 0$ | $u \leq 0$ | else |
|---|---|---|---|
| $\alpha_{L,i,m}^{(k)}$ | 1 | $\gamma$ | $\frac{u-\gamma l}{u-l}$ |
| $\alpha_{U,i,m}^{(k)}$ | 1 | $\gamma$ | $\frac{u-\gamma l}{u-l}$ |
| $\beta_{L,i,m}^{(k)}$ | 0 | 0 | 0 |
| $\beta_{U,i,m}^{(k)}$ | 0 | 0 | $\frac{(\gamma-1)ul}{u-\gamma l}$ |

**Theorem 2** (The lower bound of neural network output). *If the encoder is defined as 1, $\hat{\mathbf{A}}$ is the augmented message-passing matrix and $\sigma$ is the non-linear activation function, for $G_1 = (\mathbf{X}, \mathbf{A}_1) \in \mathcal{G}$, $\hat{\mathbf{A}} = \mathbf{D}^{-1/2}(\mathbf{A}_1 + \mathbf{I})\mathbf{D}^{-1/2} \in \hat{\mathcal{A}}$, then*

$$\underline{\tilde{f}}_{G_2,i} = \sum_{j_1 \in \mathcal{N}(i)} \hat{a}_{j_1 i}\Big[ \sum_{j_2 \in \mathcal{N}(j_1)} \hat{a}_{j_2 j_1} x_{j_2} \tilde{\mathbf{W}}_i^{(1)} + \tilde{\mathbf{b}}^{(1)}\Big] + \tilde{\mathbf{b}}^{(2)} \tag{6}$$

*where* $\quad \mathbf{W}_{2,i}^{CL} = \bar{\mathbf{z}}_{2,i} - \frac{1}{N-1}\sum_{j \neq i} \bar{\mathbf{z}}_{2,j}, \ \mathcal{N}(i) = \{j \mid a_{ij} > 0 \ or \ j = i\}, \tag{7}$

$$\lambda_{i,m}^{(2)} = \begin{cases} \alpha_{L,i,m}^{(2)} & if \ W_{2,i,m}^{CL} \geq 0 \\ \alpha_{U,i,m}^{(2)} & if \ W_{2,i,m}^{CL} < 0 \end{cases}, \ \Delta_{i,m}^{(2)} = \begin{cases} \beta_{L,i,m}^{(2)} & if \ W_{2,i,m}^{CL} \geq 0 \\ \beta_{U,i,m}^{(2)} & if \ W_{2,i,m}^{CL} < 0 \end{cases} \tag{8}$$

$$\Lambda_i^{(2)} = W_{2,i}^{CL} \odot \lambda_i^{(2)}, \ \tilde{\mathbf{W}}_i^{(\mathbf{2})} = \sum_{m=1}^{d_2} \Lambda_{i,m}^{(2)} W_{:,m}^{(2)}, \ \tilde{b}_i^{(2)} = \sum_{m=1}^{d_2} \Lambda_{i,m}^{(2)}(b_m^{(2)} + \Delta_{i,m}^{(2)}), \tag{9}$$

$$\lambda_{i,m}^{(1)} = \begin{cases} \alpha_{L,i,m}^{(1)} & if \ \tilde{W}_{i,m}^{(2)} \geq 0 \\ \alpha_{U,i,m}^{(1)} & if \ \tilde{W}_{i,m}^{(2)} < 0 \end{cases}, \ \Delta_{i,l}^{(1)} = \begin{cases} \beta_{L,i,l}^{(1)} & if \ \tilde{W}_{i,l}^{(2)} \geq 0 \\ \beta_{U,i,l}^{(1)} & if \ \tilde{W}_{i,l}^{(2)} < 0 \end{cases} \tag{10}$$

$$\mathbf{\Lambda}^{(1)} = \tilde{\mathbf{W}}_i^{(2)} \odot \lambda^{(\mathbf{1})}, \ \tilde{\mathbf{W}}_i^{(1)} = \sum_{l=1}^{d_1} \Lambda_{i,l}^{(1)}\mathbf{W}_{:,l}^{(1)}, \ \tilde{\mathbf{b}}_i^{(1)} = \sum_{l=1}^{d_2} \Lambda_{i,l}^{(1)}(b_m^{(1)} + \Delta_{i,l}^{(1)}). \tag{11}$$

*$\lambda_i^{(2)}, \mathbf{\Delta}_i^{(2)}, \mathbf{\Lambda}_i^{(2)} \in \mathbb{R}^{d_2}; \lambda_i^{(1)}, \mathbf{\Delta}_i^{(1)}, \mathbf{\Lambda}_i^{(1)} \in \mathbb{R}^{d_1}; \tilde{\mathbf{W}}_i^{(2)}, \tilde{\mathbf{W}}_i^{(1)} \in \mathbb{R}^{F \times 1}; \tilde{\mathbf{b}}^{(2)}, \tilde{\mathbf{b}}^{(1)} \in \mathbb{R}; d_1$ and $d_2$ are the dimension of the embedding of layer 1 and layer 2 respectively.*

We provide detailed proof in Appendix A.2. Theorem 2 demonstrates the process of deriving the node compactness $\underline{\tilde{f}}_{G_2,i}$ in a bound propagation manner. Before Theorem 2, we obtain all the parameters needed, and we apply the bound propagation method [32] to our problem, which is the concatenated network of the GCN encoder $f_\theta$ and a linear transformation $W_{2,i}^{CL}$. As the form of Eq. 6, the node compactness $\underline{\tilde{f}}_{G_2,i}$ can be also seen as the output of a new "GCN" encoder with network parameters $\tilde{\mathbf{W}}_i^{(\mathbf{1})}, \tilde{\mathbf{b}}^{(\mathbf{1})}, \tilde{b}^{(2)}$, which is an encoder constructed from the GCN encoder in GCL. Additionally, we present the whole process of our POT method in Algorithm 1.

**Time Complexity** The computation of the weight matrices in Theorem 2 dominates the time complexity of POT. Thus, the time complexity of our algorithm is $O(Nd_{k-1}d_k + Ed_k)$ for the $k$-th layer, which is the same as the message-passing [4, 10] in the GNN encoders.

## 5 Experiments

### 5.1 Experimental Setup

We choose four GCL baseline models for evaluation: GRACE [35], GCA [36], ProGCL [25], and COSTA [34]. Since POT is a plugin for InfoNCE loss, we integrate POT with the baselines and

**Algorithm 1:** Provable Training for GCL

---

**Input** : Graph $G = (\mathbf{X}, \mathbf{A})$, the set of all possible augmentations $\mathcal{G}$
**Output** : Node embeddings $\mathbf{Z}_1$ and $\mathbf{Z}_2$

**1** Initialize GNN encoder paramenters $\{\mathbf{W}^{(1)}, \mathbf{b}^{(1)}, \mathbf{W}^{(2)}, \mathbf{b}^{(2)}\}$;
**2** **while** *not converge* **do**
**3**   Generate graph augmentations $G_1 = (\mathbf{X}, \tilde{\mathbf{A}}_1)$ and $G_2 = (\mathbf{X}, \tilde{\mathbf{A}}_2)$;
**4**   Do forward pass, obtain $\mathbf{Z}_1 = f_\theta(\mathbf{X}, \tilde{\mathbf{A}}_1)$ and $\mathbf{Z}_2 = f_\theta(\mathbf{X}, \tilde{\mathbf{A}}_2)$, calculate $\mathcal{L}_{\text{InfoNCE}}$;
**5**   Obtain the pre-activation bounds $\{\mathbf{P}_L^{(k)}, \mathbf{P}_U^{(k)}\}$, $k = 1, 2$ as in Theorem 1;
**6**   Set $\{\alpha_L^{(k)}, \alpha_U^{(k)}, \beta_L^{(k)}, \beta_U^{(k)}\}$, $k = 1, 2$ according to Table 1;
**7**   Derive $\tilde{\underline{f}}_{G_1,i}, \tilde{\underline{f}}_{G_2,i}$ as in Theorem 2 for $i = 1$ to $N$ and calculate $\mathcal{L}_{\text{POT}}$ as in Equation 4;
**8**   Do backward propagation with $\mathcal{L} = (1 - \kappa)\mathcal{L}_{\text{InfoNCE}}(\mathbf{Z}_1, \mathbf{Z}_2) + \kappa\mathcal{L}_{\text{POT}}(G_1, G_2)$;
**9** **end**
**10** **return** $\mathbf{Z}_1, \mathbf{Z}_2$ for downstream tasks;

---

construct four models: GRACE-POT, GCA-POT, ProGCL-POT, and COSTA-POT. Additionally, we report the results of two supervised baselines: GCN and GAT [20]. For the benchmarks, we choose 8 common datasets: Cora, CiteSeer, PubMed [28], Flickr, BlogCatalog [13], Computers, Photo [18], and WikiCS [14]. Details of the datasets and baselines are presented in Appendix C.2. For datasets with a public split available [28], including Cora, CiteSeer, and PubMed, we follow the public split; For other datasets with no public split, we generate random splits, where each of the training set and validation set contains 10% nodes of the graph and the rest 80% nodes of the graph is used for testing.

## 5.2 Node Classification

Table 2: Performance ($\% \pm \sigma$) on Node Classification

| Datasets | Metrics | GCN | GAT | GRACE | | GCA | | ProGCL | | COSTA | |
|---|---|---|---|---|---|---|---|---|---|---|---|
| | | | | w/o POT | w/ POT | w/o POT | w/ POT | w/o POT | w/ POT | w/o POT | w/ POT |
| Cora | Mi-F1 | $81.9 \pm 1.1$ | $82.8 \pm 0.5$ | $78.2 \pm 0.6$ | $\mathbf{79.2 \pm 1.2}$ | $78.6 \pm 0.2$ | $\mathbf{79.6 \pm 1.4}$ | $77.9 \pm 1.4$ | $\mathbf{79.9 \pm 1.0}$ | $79.9 \pm 0.7$ | $\mathbf{80.5 \pm 0.9}$ |
| | Ma-F1 | $80.6 \pm 1.1$ | $81.8 \pm 0.3$ | $76.8 \pm 0.6$ | $\mathbf{77.8 \pm 1.4}$ | $77.7 \pm 0.1$ | $\mathbf{79.1 \pm 1.4}$ | $77.3 \pm 1.7$ | $\mathbf{78.8 \pm 0.9}$ | $79.5 \pm 0.7$ | $\mathbf{80.1 \pm 0.8}$ |
| CiteSeer | Mi-F1 | $71.6 \pm 0.6$ | $72.1 \pm 1.0$ | $66.8 \pm 1.0$ | $\mathbf{68.7 \pm 0.6}$ | $65.0 \pm 0.7$ | $\mathbf{69.0 \pm 0.8}$ | $65.9 \pm 0.9$ | $\mathbf{67.7 \pm 0.9}$ | $66.8 \pm 0.8$ | $\mathbf{67.9 \pm 0.7}$ |
| | Ma-F1 | $68.2 \pm 0.5$ | $67.1 \pm 1.3$ | $63.2 \pm 1.3$ | $\mathbf{64.0 \pm 0.9}$ | $60.8 \pm 0.8$ | $\mathbf{65.5 \pm 0.7}$ | $62.5 \pm 0.9$ | $\mathbf{63.6 \pm 0.5}$ | $63.4 \pm 1.4$ | $\mathbf{64.0 \pm 1.2}$ |
| PubMed | Mi-F1 | $79.3 \pm 0.1$ | $78.3 \pm 0.7$ | $81.6 \pm 0.5$ | $82.0 \pm 1.3$ | $80.5 \pm 2.3$ | $\mathbf{82.8 \pm 2.0}$ | $81.5 \pm 1.1$ | $81.9 \pm 0.4$ | $80.0 \pm 1.5$ | $\mathbf{80.6 \pm 1.3}$ |
| | Ma-F1 | $79.0 \pm 0.1$ | $77.6 \pm 0.8$ | $81.7 \pm 0.5$ | $\mathbf{82.4 \pm 1.2}$ | $80.5 \pm 2.1$ | $\mathbf{82.9 \pm 2.0}$ | $81.3 \pm 1.0$ | $\mathbf{82.2 \pm 0.4}$ | $79.3 \pm 1.4$ | $\mathbf{80.3 \pm 1.5}$ |
| Flickr | Mi-F1 | $48.8 \pm 1.9$ | $37.3 \pm 0.5$ | $46.5 \pm 1.8$ | $\mathbf{48.4 \pm 1.3}$ | $47.4 \pm 0.8$ | $\mathbf{50.3 \pm 0.3}$ | $46.4 \pm 1.4$ | $\mathbf{49.7 \pm 0.5}$ | $52.6 \pm 1.2$ | $\mathbf{53.8 \pm 1.2}$ |
| | Ma-F1 | $45.8 \pm 2.4$ | $35.2 \pm 0.7$ | $45.1 \pm 1.4$ | $\mathbf{47.0 \pm 1.6}$ | $46.2 \pm 0.8$ | $\mathbf{49.3 \pm 0.4}$ | $45.4 \pm 1.1$ | $\mathbf{48.9 \pm 0.3}$ | $51.2 \pm 0.9$ | $\mathbf{52.6 \pm 1.3}$ |
| BlogCatalog | Mi-F1 | $72.4 \pm 2.5$ | $68.0 \pm 2.1$ | $70.5 \pm 1.3$ | $\mathbf{72.4 \pm 1.4}$ | $76.2 \pm 0.6$ | $76.5 \pm 0.5$ | $74.1 \pm 0.6$ | $\mathbf{76.0 \pm 0.7}$ | $73.3 \pm 3.4$ | $\mathbf{75.1 \pm 1.5}$ |
| | Ma-F1 | $71.9 \pm 2.4$ | $67.6 \pm 1.8$ | $70.2 \pm 1.3$ | $\mathbf{71.8 \pm 1.4}$ | $75.8 \pm 0.7$ | $76.1 \pm 0.5$ | $73.7 \pm 0.6$ | $\mathbf{75.7 \pm 0.9}$ | $73.0 \pm 3.6$ | $\mathbf{74.9 \pm 1.6}$ |
| Computers | Mi-F1 | $87.6 \pm 0.6$ | $88.7 \pm 1.3$ | $85.9 \pm 0.4$ | $\mathbf{86.9 \pm 0.7}$ | $88.7 \pm 0.6$ | $\mathbf{89.1 \pm 0.3}$ | $88.6 \pm 0.7$ | $\mathbf{89.0 \pm 0.5}$ | $82.5 \pm 0.4$ | $82.8 \pm 0.7$ |
| | Ma-F1 | $83.1 \pm 2.3$ | $79.3 \pm 1.4$ | $83.9 \pm 0.8$ | $\mathbf{84.5 \pm 1.4}$ | $87.1 \pm 0.7$ | $\mathbf{87.6 \pm 0.5}$ | $86.9 \pm 1.1$ | $\mathbf{87.4 \pm 0.9}$ | $78.8 \pm 1.7$ | $79.0 \pm 1.0$ |
| Photo | Mi-F1 | $91.7 \pm 0.7$ | $92.0 \pm 0.5$ | $91.2 \pm 0.4$ | $\mathbf{91.8 \pm 0.3}$ | $91.5 \pm 0.4$ | $\mathbf{92.0 \pm 0.3}$ | $92.1 \pm 0.3$ | $\mathbf{92.4 \pm 0.3}$ | $91.5 \pm 0.6$ | $\mathbf{92.2 \pm 1.0}$ |
| | Ma-F1 | $88.8 \pm 2.0$ | $88.9 \pm 2.2$ | $89.2 \pm 0.7$ | $\mathbf{90.0 \pm 0.6}$ | $89.5 \pm 0.5$ | $\mathbf{90.0 \pm 0.6}$ | $90.6 \pm 0.6$ | $\mathbf{90.9 \pm 0.6}$ | $89.3 \pm 1.3$ | $\mathbf{90.1 \pm 1.4}$ |
| WikiCS | Mi-F1 | $81.2 \pm 1.2$ | $81.2 \pm 0.8$ | $80.3 \pm 0.3$ | $\mathbf{80.6 \pm 0.3}$ | $78.7 \pm 0.3$ | $\mathbf{80.7 \pm 0.3}$ | $78.8 \pm 0.4$ | $\mathbf{80.0 \pm 0.5}$ | $76.2 \pm 1.6$ | $\mathbf{79.1 \pm 0.9}$ |
| | Ma-F1 | $78.7 \pm 1.9$ | $78.5 \pm 1.8$ | $77.6 \pm 0.3$ | $\mathbf{78.0 \pm 0.3}$ | $75.4 \pm 0.1$ | $\mathbf{78.0 \pm 0.3}$ | $75.6 \pm 0.3$ | $\mathbf{77.2 \pm 0.8}$ | $72.2 \pm 1.6$ | $\mathbf{75.8 \pm 0.8}$ |

We test the performance of POT on node classification. For the evaluation process, we follow the setting in baselines [35, 36, 25, 34] as follows: the encoder is trained in a contrastive learning manner without supervision, then we obtain the node embedding and use the fixed embeddings to train and evaluate a logistic regression classifier on the downstream node classification task. We choose two measures: Micro-F1 score and Macro-F1 score. The results with error bars are reported in Table 2.

Bolded results mean that POT outperforms the baseline, with a t-test level of 0.1. As it shows, our POT method improves the performance of all the base models consistently on all datasets, which verifies the effectiveness of our model. Moreover, POT gains significant improvements over baseline models on BlogCatalog and Flickr. These two datasets are the graphs with the highest average node degree, therefore there are more possible graph augmentations in these datasets (the higher node degrees are, the more edges are dropped [35, 36]). This result illustrates our motivation that POT improves existing GCL methods when faced with complex graph augmentations.

### 5.3 Analyzing the Node Compactness

To illustrate how POT improves the training of GCL, we conduct experiments to show some interesting properties of POT by visualizing the node compactness in Definition 2. More experiments can be found in Appendix D.

**POT Improves Node Compactness.** We illustrate that our POT does improve the node compactness, which is the goal of InfoNCE. The result on Cora is shown in Figure 2. The x-axis is the epoch number, and the y-axis is the average compactness of all nodes. We can see that the node compactness of the model with POT is almost always higher than the model without POT, indicating that POT increases node compactness in the training of GCL, i.e. how the nodes follow the GCL principle is improved in the worst-case. Therefore, our POT promotes nodes to follow the GCL principle better. More results on BlogCatalog are shown in Appendix D.

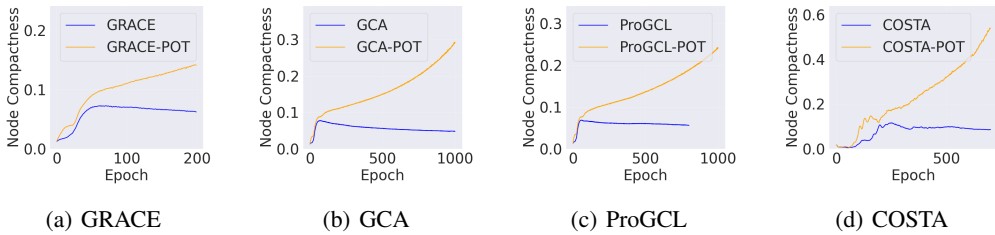

|           (a) GRACE            |           (b) GCA            |           (c) ProGCL            |           (d) COSTA            |

Figure 2: Node compactness in the training process

**Node Compactness Reflects Different Properties of Different Augmentation Strategies.** There are two types of augmentation strategies: GRACE adapts a uniform edge dropping rate, in other words, the dropping rate does not change among different nodes; GCA, however, adapts a larger dropping rate on edges with higher-degree nodes, then higher-degree nodes have a larger proportion of edges dropped. Since node compactness is how a node follows the GCL principle in the worst case, a higher degree results in larger node compactness, while a higher dropping rate results in lower node compactness because worse augmentations can be taken. To see whether node compactness can reflect those properties of different augmentation as expected, we plot the average node compactness of the nodes with the same degree as scatters in Figure 3 on BlogCatalog and Flickr. For GRACE, node degree and compactness are positively correlated, since the dropping rate does not change as the degree increases; For GCA, node compactness reflects a trade-off between dropping rate and degree: the former dominates in the low-degree stage then the latter dominates later, resulting the curve to drop then slightly rise.

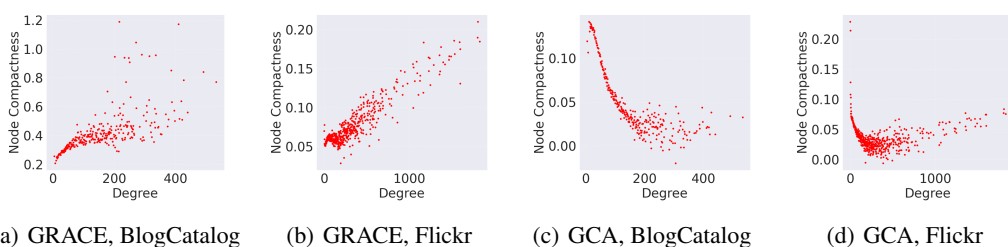

|    (a) GRACE, BlogCatalog    |    (b) GRACE, Flickr    |    (c) GCA, BlogCatalog    |    (d) GCA, Flickr    |

Figure 3: Degree and node compactness score

## 5.4 Hyperparameter Analysis

In this subsection, we investigate the sensitivity of the hyperparameter in POT, $\kappa$ in Eq. 5. In our experiments, we tune $\kappa$ ranging from $\{0.5, 0.4, 0.3, 0.2, 0.1, 0.05\}$. The hyperparameter $\kappa$ balances between InfoNCE loss and POT, i.e. the importance of POT is strengthened when $\kappa > 0.5$ while the InfoNCE is more important when $\kappa < 0.5$. We report the Micro-F1 score on node classification of BlogCatalog in Figure 4. Although the performance is sensitive to the choice of $\kappa$, the models with POT can still outperform the corresponding base models in many cases, especially on BlogCatalog where POT outperforms the base models whichever $\kappa$ is selected. The result on Cora is in Appendix D.

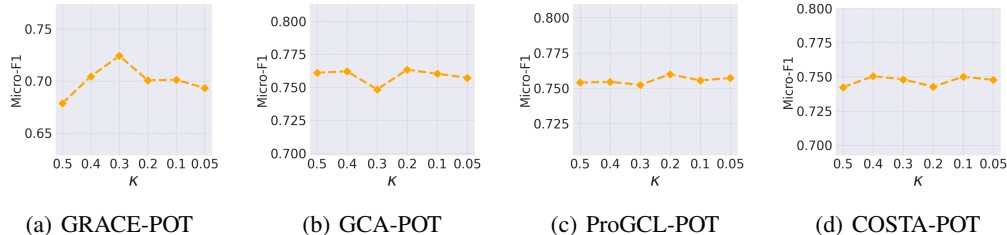

| (a) GRACE-POT | (b) GCA-POT | (c) ProGCL-POT | (d) COSTA-POT |

Figure 4: Hyperparamenter analysis on $\kappa$

## 6 Related Work

**Graph Contrastive Learning.** Graph Contrastive Learning (GCL) methods are proposed to learn the node embeddings without the supervision of labels. GRACE [35] firstly proposes GCL with random edge dropping and feature masking as data augmentations and takes InfoNCE as the objective. Based on GRACE, GCA [36] improves the data augmentation with adaptive masking and dropping rate related to node centrality. COSTA [34] proposes feature augmentation in embedding space to reduce sampling bias. ProGCL [25] reweights the negatives and empowers GCL with hard negative mining. There are also node-to-graph GCL methods, like DGI [21] and MVGRL [8]. Inspired by BYOL [6], there are also several GCL methods that do not need negative samples [19] or use BCE objectives [31].

**Bound Propagation.** Bound Propagation methods derive the output bounds directly from input bounds. Interval Bound Propagation [5] defines the input adversarial polytope as a $l_\infty$-norm ball, and it deforms as being propagated through affine layers and monotonic activations. [16] proposes a relaxation on nonlinearity via Semidefinite Programming but is restricted to one hidden layer. [24] relaxes ReLU via convex outer adversarial polytope. [32] can be applied to general activations bounded by linear functions. For GNN, [37] and [38] study the bound propagation on GNN with feature and topology perturbations respectively. One step further, [17] proposes a Mixed Integer Linear Programming based method to relax GNN certification with a shared perturbation as input. However, almost all the previous works require solving an optimization problem with some specific solver, which is time-consuming and impossible to be integrated into a training process as we expected.

## 7 Conclusion

In this paper, we investigate whether the nodes follow the GCL principle well enough in GCL training, and the answer is no. Therefore, to address this issue, we design the metric "node compactness" to measure the training of each node, and we propose POT as a regularizer to optimize node compactness, which can be plugged into the existing InfoNCE objective of GCL. Additionally, we provide a bound propagation-inspired method to derive POT theoretically. Extensive experiments and insightful visualizations verify the effectiveness of POT on various GCL methods.

**Limitations and broader impacts.** One limitation of our work is the restriction on the settings of GCL. To extend our work to more types of GCL settings and graph-level GCL, one can define the set of augmentations and network structure properly, then follow the proof in Appendix A.2 to derive the node compactness in a new setting. The detailed discussion can be found in Appendix B. We leave them for future work. Moreover, there are no negative social impacts foreseen.

## Acknowledgments and Disclosure of Funding

This work is supported in part by the National Natural Science Foundation of China (No. U20B2045, 62192784, U22B2038, 62002029, 62172052, 62322203).

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

# A Proofs and Derivations

## A.1 Proof of Theorem 1

*Proof.* We mainly make use of a conclusion in [16] and express it as the lemma below:

**Lemma 1** (Bound propagation of affine layers [16]). *In an affine layer* $\mathbf{x}^k = \mathbf{W}^{k-1}\mathbf{x}^{k-1} + \mathbf{b}^{k-1}$, *the lower bound* $\mathbf{l}^k$ *and upper bound* $\mathbf{u}^k$ *of $k$-th layer after affine transformation can be obtained by*

$$\mathbf{l}^k = [\mathbf{W}^{k-1}]_+\mathbf{l}^{k-1} + [\mathbf{W}^{k-1}]_-\mathbf{u}^{k-1} + \mathbf{b}^{k-1}, \ \mathbf{u}^k = [\mathbf{W}^{k-1}]_+\mathbf{u}^{k-1} + [\mathbf{W}^{k-1}]_-\mathbf{l}^{k-1} + \mathbf{b}^{k-1},$$

*where* $[\mathbf{W}^k]_+ = \max(\mathbf{W}^k, 0)$ *and* $[\mathbf{W}^k]_- = \min(\mathbf{W}^k, 0)$, $\mathbf{W}^k$ *is the weights of affine transformation in the $k$-th layer.*

In our problem, since $\hat{\mathbf{A}}$ is variable, we take $\mathbf{H}^{(k)}\mathbf{W}^{(k)}$ as the weights of the affine transformation and replace $\mathbf{l}^k$ and $\mathbf{u}^k$ with the bounds $\mathbf{L}$ and $\mathbf{U}$. To this end, Theorem 1 is proved. $\qquad\square$

## A.2 Proof of Theorem 2

*Proof.* Since $\mathbf{H}^{(k)}$ is the output embedding of the $k$-th layer, which implies that $\mathbf{H}^{(k-1)}$ is the input embedding of $k$-th layer. $\mathbf{P}^{(k)}$ is the pre-activation embedding in Theorem 1, $f_\theta(\mathbf{X}, A_1) = \sigma(\mathbf{P}^{(2)}), \mathbf{P}^{(2)} = \hat{\mathbf{A}}\sigma(\mathbf{P}^{(1)})\mathbf{W}^{(2)} + \mathbf{b}^{(2)}, \mathbf{P}^{(1)} = \hat{\mathbf{A}}\mathbf{X}\mathbf{W}^{(1)} + \mathbf{b}^{(1)}$. With the parameters $\alpha, \beta, \mathbf{L}, \mathbf{U}$ obtained, as well as the parameters of the GCN encoder $\{\mathbf{W}^{(1)}, \mathbf{b}^{(1)}, \mathbf{W}^{(2)}, \mathbf{b}^{(2)}\}$ given, we can derive the lower bound $\underline{\tilde{f}}_{G_2,i}$ in a bound propagation [32] manner, by taking $\hat{\mathbf{A}}$ as a variable with element-wise upper and lower bound in Definition 4:

$$\underline{\tilde{f}}_{G_2,i} = \min_{\hat{\mathbf{A}}}\{\sigma(\mathbf{P}_i^{(2)}) \cdot \mathbf{W}_{2,i}^{CL}\} \tag{12}$$

$$= \min_{\hat{\mathbf{A}}}\{\sum_{m=1}^{d_2} \sigma(\mathbf{P}_{i,m}^{(2)})W_{2,i,m}^{CL}\} \tag{13}$$

$$= \sum_{W_{2,i,m}^{CL} \geq 0} (\alpha_{L,i,m}^{(2)}(p_{i,m}^{(2)} + \beta_{L,i,m}^{(2)})) \cdot W_{2,i,m}^{CL} + \sum_{W_{2,i,m}^{CL} < 0} (\alpha_{U,i,m}^{(2)}(p_{i,m}^{(2)} + \beta_{U,i,m}^{(2)})) \cdot W_{2,i,m}^{CL} \tag{14}$$

$$= \sum_{m=1}^{d_2} W_{2,i,m}^{CL}\lambda_{i,m}^{(2)}(p_{i,m}^{(2)} + \Delta_{i,m}^{(2)}) \tag{15}$$

$$= \sum_{m=1}^{d_2} \Lambda_{i,m}^{(2)}(p_{i,m}^{(2)} + \Delta_{i,m}^{(2)}) \tag{16}$$

$$= \sum_{m=1}^{d_2} \Lambda_{i,m}^{(2)}(\sum_{j_1 \in \mathcal{N}(i)} \hat{a}_{j_1 i}\sigma(\mathbf{P}_i^{(1)})\mathbf{W}_{:,m}^{(2)} + b_m^{(2)} + \Delta_{i,m}^{(2)}) \tag{17}$$

$$= \sum_{j_1 \in \mathcal{N}(i)} \hat{a}_{j_1 i}\sigma(\mathbf{P}_i^{(1)})\tilde{\mathbf{W}}_i^{(2)} + \tilde{b}_i^{(2)} \tag{18}$$

$$= \sum_{j_1 \in \mathcal{N}(i)} \hat{a}_{j_1 i}\sum_{l=1}^{d_1} \tilde{\mathbf{W}}_{i,l}^{(2)}\lambda_{j_1,l}^{(1)}(p_{j_1,l}^{(1)} + \Delta_{j_1,l}^{(1)}) + \tilde{b}_i^{(2)} \tag{19}$$

$$= \sum_{j_1 \in \mathcal{N}(i)} \hat{a}_{j_1 i}\sum_{l=1}^{d_1} \Lambda_{i,l}^{(1)}(p_{j_1,l}^{(1)} + \Delta_{j_1,l}^{(1)}) + \tilde{b}_i^{(2)} \tag{20}$$

$$= \sum_{j_1 \in \mathcal{N}(i)} \hat{a}_{j_1 i}(\sum_{j_2 \in \mathcal{N}(j_1)} \hat{a}_{j_2 j_1}x_{j_2}\tilde{W}_i^{(1)} + \tilde{b}^{(1)}) + \tilde{b}_i^{(2)}. \tag{21}$$

(13) is the equivalent form of (12), and (14) is because: since $\alpha_{L,i,m}^{(k)}(p_{i,m}^{(k)} + \beta_{L,i,m}^{(k)}) \leq \sigma(p_{i,m}^{(k)}) \leq \alpha_{U,i,m}^{(k)}(p_{i,m}^{(k)} + \beta_{U,i,m}^{(k)})$, then if $W_{2,i,m}^{CL}$ is multiplied to the inequality, we have

$$\alpha_{L,i,m}^{(k)}(p_{i,m}^{(k)} + \beta_{L,i,m}^{(k)})W_{2,i,m}^{CL} \leq \sigma(p_{i,m}^{(k)})W_{2,i,m}^{CL} \leq \alpha_{U,i,m}^{(k)}(p_{i,m}^{(k)} + \beta_{U,i,m}^{(k)})W_{2,i,m}^{CL} \quad \text{if } W_{2,i,m}^{CL} \geq 0,$$
(22)

$$\alpha_{U,i,m}^{(k)}(p_{i,m}^{(k)} + \beta_{U,i,m}^{(k)})W_{2,i,m}^{CL} \leq \sigma(p_{i,m}^{(k)})W_{2,i,m}^{CL} \leq \alpha_{L,i,m}^{(k)}(p_{i,m}^{(k)} + \beta_{L,i,m}^{(k)})W_{2,i,m}^{CL} \quad \text{if } W_{2,i,m}^{CL} < 0.$$
(23)

In (15), we combine $\alpha_L^{(2)}$ and $\alpha_U^{(2)}$ into a single parameter $\lambda_i^{(2)}$, and similarly for $\Delta_i^{(2)}$. In (16), $\lambda_i^{(2)}$ and $\mathbf{W}_{2,i}^{CL}$ are combined to be $\Lambda_i^{(2)}$. We expand the expression of $p_{i,m}^{(2)}$ to get (17). Since $m$ is not related to $j_1$, we move the outer sum into the inner sum, and the parameters can be further combined, therefore we get (18). With the ideas above, the following equations can be derived, and finally, we get (21) which is exactly the form in Theorem 2. □

## B  A Detailed Discussion on the Limitations

**The definition of augmented adjacency matrix.** In Definition 3, we present the definitions and theorems under the condition of edge dropping since it is the most common topology augmentation in GCL [35, 31, 19]. The proposed POT is suitable for various augmentation strategies including edge dropping and adding. POT is applicable as long as the range of augmentation is well-defined in Definition 3. For example, if edge addition is allowed, the first constraint in Definition 3 can be removed, resulting in the change of the element-wise bound in Definition 5. After the element-wise bound is defined, the other computation steps are the same, then the POT loss with edge addition can be derived. Since edge addition is not a common practice and all the baselines use edge dropping only, we set the first constraint in Definition 3 to obtain a tighter bound.

**Applying POT to graph-level GCL.** If the downstream task requires graph embeddings, POT still stands. First, related concepts like "graph compactness" can be defined similarly to node compactness. Second, the pooling operators like "MEAN", "SUM", and "MAX" are linear, therefore the form of "graph compactness" can be derived with some modifications on the steps in Appendix A.2.

**Generalizing to Other Types of Encoders.** Definition 5 and Theorem 2 are based on the assumption that the encoder is GCN since GCN is the most common choice in GCL. However, other encoders are possible to be equipped with POT. GraphSAGE [7] has a similar form to GCN, then the derivation steps are similar. For Graph Isomorphism Network [26] and its form is

$$h_v^{(k+1)} = \text{MLP}((1 + \epsilon)h_v^{(k)} + \sum_{u \in \mathcal{N}(v)} h_u^{(k)}).$$

We can first relax the nonlinear activation functions in MLP with Definition 5, then obtain the pre-activation bounds with Theorem 1, and finally follow the steps in Appendix A.2 to derive the form of node compactness when a GIN encoder is applied.

**The provable training of feature augmentation** In this paper, we focus on topology augmentations. However, as we have mentioned in the limitation, feature augmentation is also a common class of augmentations. We explore the provable training for feature augmentation as follows. Unlike topology augmentation, the augmented feature is only the input of the first layer, so it is easier to deal with. Inspired by [37], we can relax the discrete manipulation into $l - 1$ constraints, then the node compactness is related to the feasible solution of the dual form. A similar binary cross-entropy loss can be designed. More detailed derivations are still working in progress.

However, more efforts are still to be made to deal with some subtle parts. Future works are in process.

## C  Experimental Details

### C.1  Source Code

The complete implementation can be found at https://github.com/VoidHaruhi/POT-GCL. We also provide an implementation based on GammaGL [12] at https://github.com/BUPT-GAMMA/GammaGL.

## C.2 Datasets and Baselines

We choose 8 common datasets for evaluations: Cora, CiteSeer, and PubMed[28], which are the most common graph datasets, where nodes are paper and edges are citations; BlogCatalog and Flickr, which are social network datasets proposed in [13], where nodes represent users and edges represent social connections; Computers and Photo [18], which are Amazon co-purchase datasets with products as nodes and co-purchase relations as edges; WikiCS [14], which is a graph dataset based on Wikipedia in Computer Science domain. The statistics of the datasets are listed in Table 3. A percentage in the last 3 columns means the percentage of nodes used in the training/validation/testing phase.

Table 3: The statistics of datasets

| Datasets | Nodes | Edges | Features | Classes | Training | Validation | Test |
|---|---|---|---|---|---|---|---|
| Cora | 2708 | 10556 | 1433 | 7 | 140 | 500 | 1000 |
| CiteSeer | 3327 | 9104 | 3703 | 6 | 120 | 500 | 1000 |
| PubMed | 19717 | 88648 | 500 | 3 | 60 | 500 | 1000 |
| BlogCatalog | 5196 | 343486 | 8189 | 6 | 10% | 10% | 80% |
| Flickr | 7575 | 479476 | 12047 | 9 | 10% | 10% | 80% |
| Computers | 13752 | 491722 | 767 | 10 | 10% | 10% | 80% |
| Photo | 7650 | 238162 | 645 | 8 | 10% | 10% | 80% |
| WikiCS | 11701 | 216123 | 300 | 10 | 10% | 10% | 80% |

We obtain the datasets from PyG [3]. Although the datasets are available for public use, we cannot find their licenses. The datasets can be found in the URLs below:

- Cora, CiteSeer, PubMed: https://github.com/kimiyoung/planetoid/raw/master/data
- BlogCatalog: https://docs.google.com/uc?export=download&id=178PqGqh67RUYMMP6-SoRHDoIBh8ku5FS&confirm=t
- Flickr: https://docs.google.com/uc?export=download&id=1tZp3EB20fAC27SYWwa-x66_8uGsuU62X&confirm=t
- Computers, Photo: https://github.com/shchur/gnn-benchmark/raw/master/data/npz/
- WikiCS: https://github.com/pmernyei/wiki-cs-dataset/raw/master/dataset

For the implementations of baselines, we use PyG to implement GCN and GAT. For other GCL baselines, we use their original code. The sources are listed as follows:

- GCN: https://github.com/pyg-team/pytorch_geometric/blob/master/examples/gcn.py
- GAT: https://github.com/pyg-team/pytorch_geometric/blob/master/examples/gat.py
- GRACE: https://github.com/CRIPAC-DIG/GRACE
- GCA: https://github.com/CRIPAC-DIG/GCA
- ProGCL: https://github.com/junxia97/ProGCL
- COSTA: https://github.com/yifeiacc/COSTA

We implement our POT based on the GCL baselines with PyTorch.

## C.3 Hyperparameter Settings

To make a fair comparison, we set the drop rate to be consistent when testing the performance gained by our proposed POT. The drop rates of edges $(p_e^1, p_e^2)$ are given in Table 4:

Table 4: Hyperparameters: $(p_e^1, p_e^2)$

| Models | Cora | CiteSeer | PubMed | Flickr | BlogCatalog | Computers | Photo | WikiCS |
|---|---|---|---|---|---|---|---|---|
| GRACE | (0.4, 0.3) | (0.3, 0.1) | (0.4, 0.1) | (0.4, 0.1) | (0.4, 0.3) | (0.6, 0.4) | (0.5, 0.3) | (0.2, 0.3) |
| GCA | (0.3, 0.1) | (0.2, 0.2) | (0.4, 0.1) | (0.3, 0.1) | (0.3, 0.1) | (0.6, 0.3) | (0.3, 0.5) | (0.2, 0.3) |
| ProGCL | (0.3, 0.1) | (0.4, 0.3) | (0.4, 0.1) | (0.4, 0.1) | (0.4, 0.2) | (0.6, 0.3) | (0.6, 0.3) | (0.2, 0.3) |
| COSTA | (0.2, 0.4) | (0.3, 0.1) | (0.4, 0.1) | (0.4, 0.1) | (0.4, 0.2) | (0.6, 0.3) | (0.3, 0.5) | (0.2, 0.3) |

Additionally, we provide the values of the hyperparameter $\kappa$ in our POT, as well as the temperature $\tau$, in Table 5:

Table 5: Hyperparameters: $(\tau, \kappa)$

| Models | Cora | CiteSeer | PubMed | Flickr | BlogCatalog | Computers | Photo | WikiCS |
|---|---|---|---|---|---|---|---|---|
| GRACE-POT | (0.7, 0.4) | (0.9, 0.5) | (0.5, 0.1) | (0.3, 0.1) | (0.5, 0.3) | (0.3, 0.1) | (0.2, 0.4) | (0.3, 0.05) |
| GCA-POT | (0.6, 0.3) | (0.7, 0.5) | (0.3, 0.1) | (0.2, 0.5) | (0.3, 0.3) | (0.2, 0.2) | (0.6, 0.2) | (0.3, 0.1) |
| ProGCL-POT | (0.4, 0.4) | (0.5, 0.2) | (0.5, 0.3) | (0.3, 0.5) | (0.3, 0.2) | (0.3, 0.2) | (0.2, 0.3) | (0.4, 0.1) |
| COSTA-POT | (0.4, 0.1) | (0.3, 0.05) | (0.4, 0.3) | (0.2, 0.1) | (0.05, 0.4) | (0.1, 0.1) | (0.2, 0.3) | (0.4, 0.3) |

We provide more hyperparameters which may also have effects on the performance. In Table 6, "num_epochs" is the epoch number for training, and "pot_batch" is the batch size of POT, "-1" means full batch is used.

Table 6: Hyperparameters: (pot_batch, num_epochs)

| Models | Cora | CiteSeer | PubMed | Flickr | BlogCatalog | Computers | Photo | WikiCS |
|---|---|---|---|---|---|---|---|---|
| GRACE-POT | (-1, 200) | (-1, 200) | (256, 2500) | (512, 1000) | (-1, 1000) | (512, 2500) | (512, 0.4) | (512, 4000) |
| GCA-POT | (-1, 700) | (-1, 1000) | (256, 2500) | (1024, 2500) | (1024, 1500) | (256, 2500) | (1024, 2500) | (1024, 3500) |
| ProGCL-POT | (-1, 1000) | (-1, 700) | (256, 2000) | (512, 2000) | (1024, 1500) | (256, 3000) | (1024, 2500) | (256, 3000) |
| COSTA-POT | (-1, 600) | (-1, 1500) | (512, 2000) | (1024, 2500) | (1024, 2500) | (256, 2500) | (512, 3500) | (256, 3000) |

## C.4 Environment

The environment where our codes run is as follows:

- OS: Linux 5.4.0-131-generic

- CPU: Intel(R) Xeon(R) Gold 6348 CPU @ 2.60GHz

- GPU: GeForce RTX 3090

# D More Experimental Results

## D.1 More Results for Section 3

We present more results of the imbalance of GCL training on Cora and Flickr shown in Figure 5. The conclusion is similar: it can be observed that the training of GCL on more datasets is also imbalanced on the untrained nodes.

## D.2 More Results for Section 5.3

We provide more results of the experiments in Section 5.3: the node compactness score during training on BlogCatalog in Figure 6. From the results, we can see that the conclusion in Section 5.3 is the same on more datasets: our POT can promote node compactness compared to the original GCL methods, which can enforce the nodes to follow the GCL principle better.

## D.3 More Results of Hyperparameter Analysis

The result of the hyperparameter analysis of $\kappa$ on Cora is shown in Figure 7. POT outperforms the baseline models with most of the choices of $\kappa$, which implies that the proposed POT is robust to the change of $\kappa$.

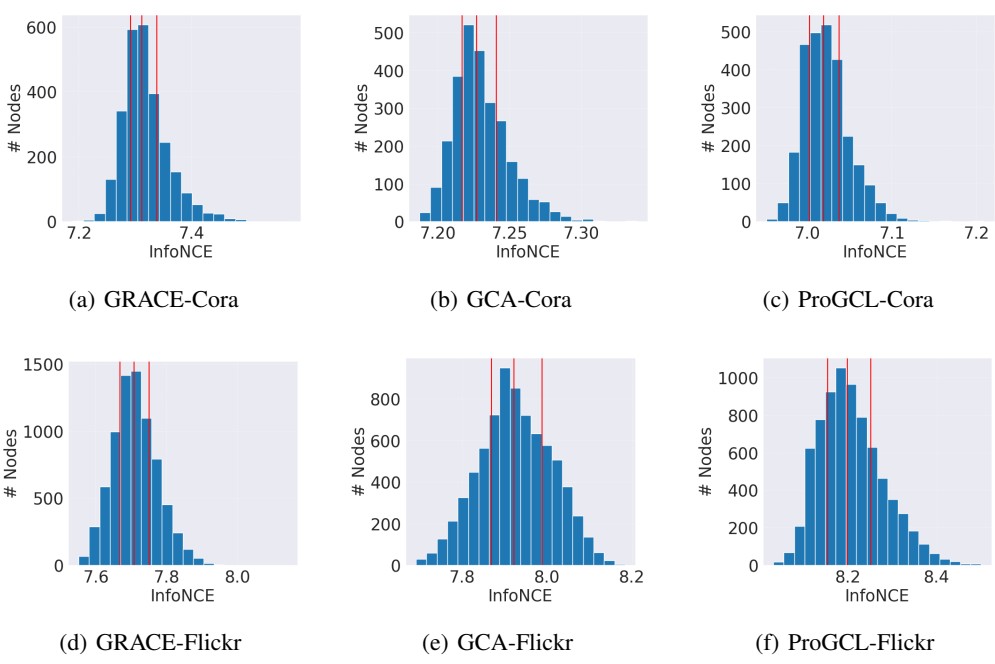

Figure 5: More results showing the imbalance of GCL training

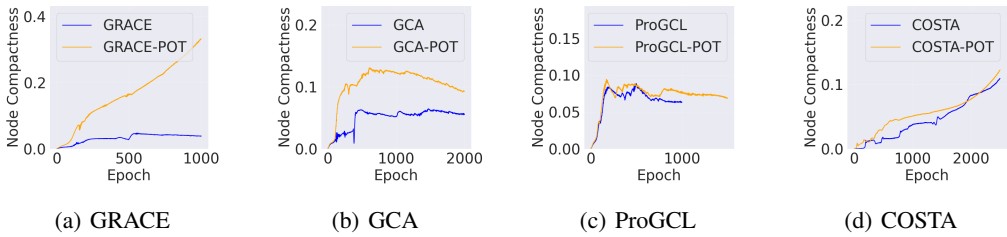

Figure 6: Node compactness in the training process on BlogCatalog

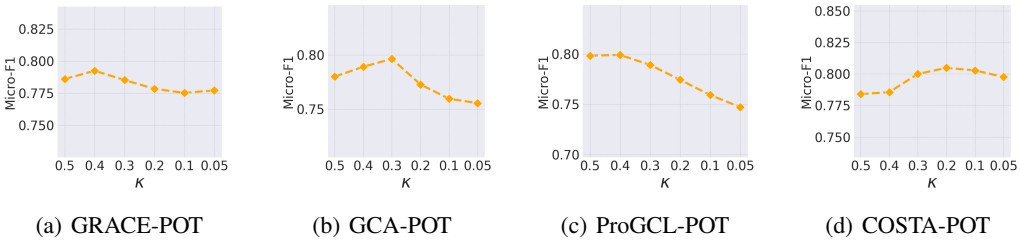

Figure 7: Hyperparamenter analysis of $\kappa$ on Cora

## D.4 More Insights and Explanations of Experiments on Node Compactness

**Node Compactness during training.** In Figure 2 and Figure 6, we observe that the node compactness of traditional GCL methods may decreases with the increase of epoch. We think the "false negative" issue may contribute to the drop in the curves in Figure 2. Since GCL uses all $N-1$ nodes as negative samples, there are many false negatives, i.e., some nodes with the same class as the anchor node are treated as negative samples in the contrastive loss. As in the definition, the node compactness measures the worst case of how well a node behaves across all possible augmentations. Therefore, as

the epoch increases and some false negative information is captured, that "worst case" value may decrease. As a result, we observe a drop in node compactness in traditional GCL methods. There are some existing methods alleviating this, called "hard negative mining", including the baseline method ProGCL. From Figure 2, we see that the curve of ProGCL almost does not drop, while the average node compactness of other methods decreases. Since our POT method explicitly optimizes the node compactness, the curve will continue to rise as the epoch increases. It is an open question and may be related to fundamental issues in contrastive learning.

**Understanding the result in Figure 3.** First, edges are randomly dropped with a uniform rate in GRACE. Since there are fewer unimportant edges around a low-degree node, those informative edges are more easily dropped, which prevents the contrastive loss from learning useful semantics, therefore those low-degree nodes are hard to be well-trained in GRACE. This is also the motivation of GCA. To alleviate this, GCA sets the dropping rate of edges by node centrality, keeping important edges around low-degree nodes. That's why we observe a high node compactness in low-degree nodes. However, the augmentation strategy of GCA puts a higher dropping rate for high-degree nodes. This may explain why node compactness drops as the degree increases at first. When the degree comes to relatively large, the node is "strong" enough for this augmentation strategy and is prone to be well-trained. To conclude, the node degree and node compactness are positively correlated in GRACE, however, there is a trade-off between degree and node compactness in GCA. This may explain why the result in Figure 3 is reasonable.

### D.5 More Experimental Results

**POT can improve the InfoNCE loss.** In Section 2, we investigate the problem of existing GCL training with InfoNCE, since it is the only metric we have. After that, we propose "node compactness" as a more appropriate and intrinsic metric to evaluate the training of a node in GCL. Therefore, we conduct experiments in the Figures to show the validity of node compactness as well as some properties. That is the reason why we did not provide further analysis of InfoNCE. However, it is still inspiring to investigate whether POT can improve InfoNCE. Intuitively, POT can improve the InfoNCE loss. POT has a larger regularization on the nodes that are not well-trained across all possible augmentations, as a result, the training of nodes under two specific augmentations is also improved. To illustrate this, we conduct an experiment comparing the InfoNCE loss values of the nodes with/without using POT, similar to the experiment in Section 2. We choose GRACE and GCA as baselines and show the result of Cora in Figure 8. It can be seen that InfoNCE is improved by POT in two aspects: the average InfoNCE of the nodes is reduced, and the distribution of InfoNCE values becomes more centralized and balanced.

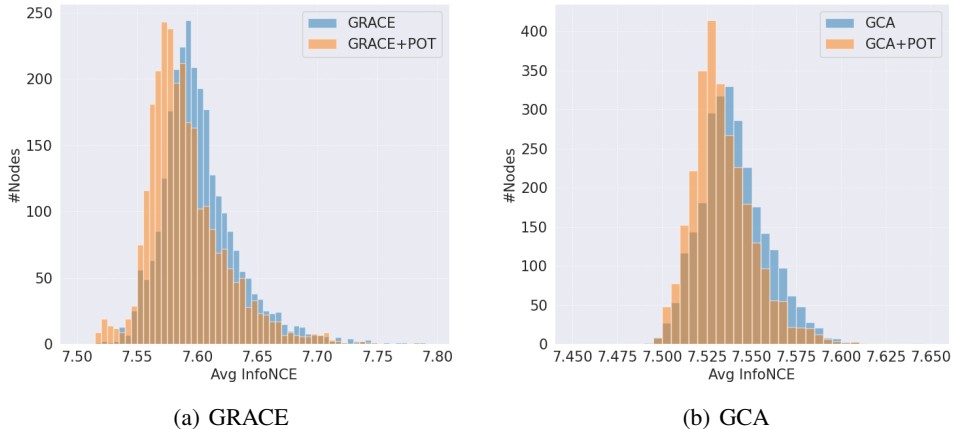

(a) GRACE          (b) GCA

Figure 8: POT improves InfoNCE. The average of InfoNCE values is reduced and the loss values are more balanced.

**The applicability of POT to different augmentation strategies.** To investigate whether our POT applies to various graph augmentations, we evaluate the performance of POT under four types of topology augmentation: random edge dropping proposed in GRACE; node centrality-based topology

augmentations proposed in GCA, including degree centrality, eigenvector centrality, and PageRank centrality. The backbone of the GCL baseline is GRACE. The result on Cora and Flickr is given in Table 7. POT consistently outperforms the baseline models with different graph augmentations. As stated in the Limitation section, since more theoretical verifications are needed, POT of other augmentations, including feature augmentations, are working in progress. We are looking forward to more capability of POT.

Table 7: POT improves existing GCL with different augmentations (Random: random edge dropping with a uniform rate; Degree: degree-centrality dropping rate; Eigenvector: eigenvector-centrality dropping rate; PageRank: PageRank-centrality dropping rate).

| Dataset | Method | Random | | Degree | | Eigenvector | | PageRank | |
|---|---|---|---|---|---|---|---|---|---|
| | | F1Mi | F1Ma | F1Mi | F1Ma | F1Mi | F1Ma | F1Mi | F1Ma |
| Cora | GCL | $78.2 \pm 0.6$ | $76.8 \pm 0.6$ | $78.6 \pm 0.2$ | $77.7 \pm 0.1$ | $78.0 \pm 0.8$ | $77.3 \pm 0.6$ | $77.8 \pm 1.1$ | $77.0 \pm 1.2$ |
| | GCL+POT | $\mathbf{79.2 \pm 1.2}$ | $\mathbf{77.8 \pm 1.4}$ | $\mathbf{79.6 \pm 1.4}$ | $\mathbf{79.1 \pm 1.4}$ | $\mathbf{79.1 \pm 1.2}$ | $\mathbf{78.3 \pm 1.5}$ | $\mathbf{79.1 \pm 1.0}$ | $\mathbf{77.8 \pm 1.4}$ |
| Flickr | GCL | $46.5 \pm 1.8$ | $45.1 \pm 1.4$ | $47.4 \pm 0.8$ | $46.2 \pm 0.8$ | $47.7 \pm 1.2$ | $46.6 \pm 1.3$ | $48.4 \pm 1.0$ | $47.3 \pm 1.2$ |
| | GCL+POT | $\mathbf{49.8 \pm 1.0}$ | $\mathbf{48.7 \pm 0.8}$ | $\mathbf{50.3 \pm 0.3}$ | $\mathbf{49.3 \pm 0.4}$ | $\mathbf{49.7 \pm 1.0}$ | $\mathbf{48.8 \pm 1.1}$ | $\mathbf{50.2 \pm 0.8}$ | $\mathbf{49.3 \pm 1.0}$ |

**Node compactness can reflect the nodes' sensitivity to augmentations.** We conduct an experiment to show the relationship between a node's compactness and the standard error of its InfoNCE loss under different graph augmentations. Specifically, we first train a GCL model, then fix the network parameters and sample 500 pairs of different augmentations. After encoding these augmentations, 500 InfoNCE loss values are obtained for each node. We choose to calculate the standard error of these 500 values of each node to reflect its sensitivity to different augmentations. We also compute each node's compactness as Theorem 2 in the paper, which is not related to specific augmentations. With the process above, we have a data sample (node_compactness, InfoNCE_std) for each node. We divide these samples into several bins by the value of node compactness and calculate the average of InfoNCE_std in each bin. Finally, we show the result as a scatter plot in Figure 9. It can be seen that less sensitive nodes have higher node compactness values. To conclude, this experiment shows that our proposed node compactness can be a proxy of the node's sensitivity to different graph augmentations, as the motivation stated.

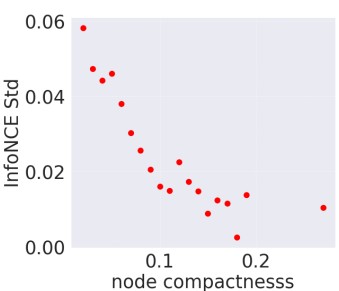

Figure 9: Node compactness and the standard error of InfoNCE loss under different augmentations. More sensitive nodes have lower node compactness as expected.

