# OpenReview forum: "Provable Training for Graph Contrastive Learning"
_NeurIPS.cc/2023/Conference — NeurIPS 2023 spotlight_

### Official Review · Reviewer_AXri · 2023-07-05

**Soundness:** 3 good
**Presentation:** 3 good
**Contribution:** 4 excellent
**Rating:** 8
**Confidence:** 4

**Summary:**

The goal of this paper is to investigate the properties of different nodes in GCL with different graph augmentations. The paper discovers the imbalanced training issue of GCL methods, and proposes the concept “node compactness”, measuring how each node follows the GCL principle. Finally, the paper proposes the node compactness regularization, and shows its effectiveness by combining it with existing GCL methods.

**Strengths:**

- The paper is well motivated with both theoretical and empirical analysis.
- The paper provides some interesting insights on the node properties in GCL.
- The effectiveness of the proposed model is well demonstrated by the empirical results.

**Weaknesses:**

- One of the motivations is that “how to distinguish these nodes”, while it seems that the paper doesn’t distinguish these nodes?
- Although the presentation is good enough, I’m not very clear with the whole training process because of the math-heavy part in the technique part. Do we need to compute the POT first, and then train InfoNCE loss? Or do we need to train the model with the two steps iteratively?
- In theorem 1, the conclusion is obtained under the condition that AXW+b, so if the GCN process is not AXW+b, can we still use POT regularization?

**Questions:**

See Weaknesses

**Limitations:**

The discussion section of the paper is concise but unsatisfactory in addressing the main points.

---

> ### Author Rebuttal · Authors · 2023-08-09
>
> We sincerely appreciate the positive comments and valuable feedback from the reviewer. Below, we address the reviewer's concerns one by one, hoping that a better understanding of every point can be delivered.
> 1. > One of the motivations is that “how to distinguish these nodes”, while it seems that the paper doesn’t distinguish these nodes?
>
>     **Response**: Thanks. Specifically, the value of node compactness serves as a good metric to distinguish all the nodes in a graph by how much they are prone to be well-trained. It is a relative measure, instead of giving an explicit bound of discrimination. In Figure 3, nodes with different properties are distinguished by the node compactness, directly addressing this motivation.
>
> 2. > Although the presentation is good enough, I’m not very clear with the whole training process because of the math-heavy part in the technique part. Do we need to compute the POT first, and then train InfoNCE loss? Or do we need to train the model with the two steps iteratively?
>
>     **Response**: Sorry for the confusion. We describe the whole process of graph contrastive learning with the provable training as follows:
>
>     (1) Generate two augmented graphs $G_1$ and $G_2$ from $G$;
>
>     (2) Perform the forward pass and the node embeddings $Z_1$ and $Z_2$ are obtained. Compute the InfoNCE loss $\mathcal L_{\text{InfoNCE}}(Z_1. Z_2)$;
>
>     (3) Compute the POT loss as Algorithm 1 in our paper described;
>
>     (4) Combine the two losses as in Equation 5, then do the backward propagation to update the network parameters;
>
>     (5) Iteratively do steps (1)-(4) until stop.
>
>    In these steps, step (1)-(2) is the same as traditional GCL. We will improve the presentation of Algorithm 1 as above in the revision.
> 3. > In theorem 1, the conclusion is obtained under the condition that AXW+b, so if the GCN process is not AXW+b, can we still use POT regularization?
>
>     **Response**: Yes, POT is capable of various graph encoders. Since GCN is the most common encoder in GCL methods, we mainly provided the derivation of the form of node compactness for GCN. For other types of encoders, the form of node compactness can be derived similarly using Definition 5 and Theorem 1. For example, if the encoder is Graph Isomorphism Network, and its form is
>     $$
>     h_v^{(k+1)}=\text{MLP}((1+\epsilon)h_v^{(k)}+\sum_{u\in \mathcal N(v)}h_u^{(k)}).
>     $$
>     We can first relax the nonlinear activation functions in MLP with Definition 5, then obtain the pre-activation bounds with Theorem 1, and finally follow the steps in Appendix A.2 to derive the form of node compactness when a GIN encoder is applied.
>     We will add this discussion in the revision to improve the "Limitation" section.
> 4. > The discussion section of the paper is concise but unsatisfactory in addressing the main points.
>
>    **Response**: We have expanded the discussion of limitations. Please refer to the "global" rebuttal.

---

> > ### Comment · Reviewer_AXri · 2023-08-21
> >
> > Thanks for the detailed response and clairification. I appreciate the authors' effort, which have addressed my concerns and corrected my misunderstanding. I'm willing to increase my rating.

---

### Official Review · Reviewer_Kesd · 2023-07-05

**Soundness:** 3 good
**Presentation:** 3 good
**Contribution:** 4 excellent
**Rating:** 7
**Confidence:** 5

**Summary:**

The paper considers an important problem in graph contrastive learning, i.e., the relationship between the node property with the graph augmentations. The main takeaway is that the training of GCL on different nodes is imbalanced, and the concept “node compactness” is introduced to guarantee the training of GCL is provable. This is demonstrated on eight benchmarks.

**Strengths:**

S1. The paper topic is interesting.
S2. The theoretical foundation is solid.
S3. The paper is relatively well-written and well-organized.

**Weaknesses:**

The experiments section could be strengthened. I’m not sure that whether the following experiment needs to be conducted. Specifically, the paper aims to discover the nodes that are not sensitive to different graph augmentations. However, there is no experiment to analyze this, i.e., maybe we can generate different graph augmentations to check whether some nodes are always well trained, or not well trained? Moreover, some notations are not described, e.g., [ ]+ and [ ]- in Theorem 1. Also, why the curve of traditional GCL methods decreases with the increase of Epoch in Fig.2?

**Questions:**

See the above weakness.

---

> ### Author Rebuttal · Authors · 2023-08-09
>
> We sincerely thank the reviewer for the precious time spent reading through the paper and giving constructive suggestions. To address the concerns, we clarify the experiment section and some notations as follows.
> 1. > The experiments section could be strengthened. I’m not sure whether the following experiment needs to be conducted. Specifically, the paper aims to discover the nodes that are not sensitive to different graph augmentations. However, there is no experiment to analyze this, i.e., maybe we can generate different graph augmentations to check whether some nodes are always well trained, or not well trained?
>
>     **Response**: We greatly thank the reviewer for this suggestion. To investigate whether our proposed node compactness can reflect the nodes' sensitivity to augmentations, we conduct an experiment to show the relationship between a node's compactness and the standard error of its InfoNCE loss under different graph augmentations. Specifically, we first train a GCL model, then fix the network parameters and sample 500 pairs of different augmentations. After encoding these augmentations, 500 InfoNCE loss values are obtained for each node. We choose to calculate the standard error of these 500 values of each node to reflect its sensitivity to different augmentations. We also compute each node's compactness as Theorem 2 in the paper, which is not related to specific augmentations. With the process above, we have a data sample (node_compactness, InfoNCE_std) for each node. We divide these samples into several bins by the value of node compactness and calculate the average of InfoNCE_std in each bin. Finally, we show the result as a scatter plot in Figure 2 in the rebuttal PDF. It can be seen that less sensitive nodes have higher node compactness values. To conclude, this experiment shows that our proposed node compactness can be a proxy of the node's sensitivity to different graph augmentations, as the motivation stated. I hope this experiment can make the paper more self-contained.
> 2. > Moreover, some notations are not described, e.g., [ ]+ and [ ]- in Theorem 1.
>
>    **Response**: Sorry for the confusion in the notations. Here $[X]\_+$ denotes $\max(X,0)$ and $[X]\_-$ denotes $\min(X,0)$. We will check the description of the notations in the revision thoroughly.
>
> 3. > Also, why the curve of traditional GCL methods decreases with the increase of Epoch in Fig.2?
>
>    **Response**: Thanks for pointing this out. We think the "false negative" issue may contribute to the drop in the curves in Figure 2. Since GCL uses all $N-1$ nodes as negative samples, there are many false negatives, i.e., some nodes with the same class as the anchor node are treated as negative samples in the contrastive loss. As in the definition, the node compactness measures the worst case of how well a node behaves across all possible augmentations. Therefore, as the epoch increases and some false negative information is captured, that "worst case" value may decrease. As a result, we observe a drop in node compactness in traditional GCL methods.
>    There are some existing methods alleviating this, called "hard negative mining", including the baseline method ProGCL. From Figure 2, we see that the curve of ProGCL almost does not drop, while the average node compactness of other methods decreases. Since our POT method explicitly optimizes the node compactness, the curve will continue to rise as the epoch increases.
>    It is an open question and may be related to fundamental issues in contrastive learning. We look forward to further discussions.

---

> > ### Comment · Reviewer_Kesd · 2023-08-19
> > **Thanks for your response**
> >
> > Thanks for your response. I have no further questions.

---

### Official Review · Reviewer_QmjC · 2023-07-06

**Soundness:** 4 excellent
**Presentation:** 3 good
**Contribution:** 4 excellent
**Rating:** 7
**Confidence:** 4

**Summary:**

Graph augmentation is a fundamental component for graph contrastive learning. When augmenting graph structures, how the change of structures affects the GCL is an interesting problem. In this work, the paper proposes the “node compactness” to describe the behavior of different nodes, i.e., whether there are some nodes consistently stable enough to the training process with different graph augmentations. With this concept, the paper designs a new POT regularization term as a plug-in, which enables the training process of nodes to follow the GCL principle, so as to improve the performance of GCL finally.

**Strengths:**

1.	The paper studies an important and challenging problem. The idea of node compactness is novel.
2.	Using bound propagation to derive node compactness in GCL is also interesting and technically sound.
3.	Thorough experiments are conducted to validate how the proposed model works.


**Weaknesses:**

1.	Some techniques are not introduced clearly (see comments below).
2.	Some experimental results are also not clearly explained (see comments below).


**Questions:**

1.	The first constraint in Definition 3 requires the graph augmentation is edge dropping. Does that mean the proposed provable training is only suitable for edge dropping?
2.	The experiment analyzes the relationship between the node compactness with the properties of different augmentation strategies, while some statements are not well explained, which may be hard to understand. Specifically, why does a higher degree result in larger node compactness? Why the results in Fig.3 are reasonable? More details should be provided.

---

> ### Author Rebuttal · Authors · 2023-08-09
>
> We greatly thank the reviewer for your interest in our paper and constructive suggestions. To make further clarification on the techniques and experimental details, we respond to the reviewer's questions one by one. We look forward to assisting to have a better understanding of our paper.
> 1. > The first constraint in Definition 3 requires the graph augmentation is edge dropping. Does that mean the proposed provable training is only suitable for edge dropping?
>
>     **Response**: Thanks. We present the definitions and theorems under the condition of edge dropping since it is the most common topology augmentation in GCL[1, 2, 3]. The proposed POT is suitable for various augmentation strategies including edge dropping and adding. POT is applicable as long as the range of augmentation is well-defined in Definition 4. For example, if edge addition is allowed, the first constraint in Definition 3 can be removed, resulting in the change of the element-wise bound in Definition 4. After the element-wise bound is defined, the other computation steps are the same, then the POT loss with edge addition can be derived. Since edge addition is not a common practice and all the baselines use edge dropping only, we set the first constraint in Definition 3 to obtain a tighter bound. This discussion will be added to the revision.
>
> 2. > The experiment analyzes the relationship between the node compactness with the properties of different augmentation strategies, while some statements are not well explained, which may be hard to understand. Specifically, why does a higher degree result in larger node compactness? Why the results in Fig.3 are reasonable? More details should be provided.
>
>     **Response**: Sorry for the confusion. First, edges are randomly dropped with a uniform rate in GRACE. Since there are fewer unimportant edges around a low-degree node, those informative edges are more easily dropped, which prevents the contrastive loss from learning useful semantics, therefore those low-degree nodes are hard to be well-trained in GRACE. This is also the motivation of GCA. To alleviate this, GCA sets the dropping rate of edges by node centrality, keeping important edges around low-degree nodes. That's why we observe a high node compactness in low-degree nodes. However, the augmentation strategy of GCA puts a higher dropping rate for high-degree nodes. This may explain why node compactness drops as the degree increases at first. When the degree comes to relatively large, the node is "strong" enough for this augmentation strategy and is prone to be well-trained.
>     To conclude, the node degree and node compactness are positively correlated in GRACE, however, there is a trade-off between degree and node compactness in GCA. This may explain why the result in Figure 3 is reasonable.
>
> References:
>
> [1] Zhu, Yanqiao, et al. "Deep graph contrastive representation learning." arXiv preprint arXiv:2006.04131 (2020).
>
> [2] Zhang, Hengrui, et al. "From canonical correlation analysis to self-supervised graph neural networks." Advances in Neural Information Processing Systems 34 (2021): 76-89.
>
> [3] Thakoor, Shantanu, et al. "Bootstrapped representation learning on graphs." ICLR 2021 Workshop on Geometrical and Topological Representation Learning. 2021.

---

### Official Review · Reviewer_ynwF · 2023-07-07

**Soundness:** 4 excellent
**Presentation:** 3 good
**Contribution:** 4 excellent
**Rating:** 7
**Confidence:** 5

**Summary:**

The paper aims at studying the node properties given different graph augmentations in graph contrastive learning. It has the following contributions. 1) It discovers the training of GCL methods is severely imbalanced. 2) It proposes a novel concept of “node compactness”, and the provable training for GCL with the concept. 3) Besides the theoretical analysis, the paper uses extensive numerical results to show the effectiveness of the proposed method. Generally, the paper is interesting and easy to follow.

**Strengths:**

- The paper addresses an important problem in the GCL community and is engaging to read.
- The technical contribution of the paper is novel and brings a fresh perspective to the field.
- The proposed model is thoroughly evaluated on various benchmarks, demonstrating its effectiveness and providing strong evidence for its performance.


**Weaknesses:**

- The experiment section may lack some essential components or details, and therefore it could be considered insufficient.
- The discussion on the limitations of the proposed method is relatively brief. It would be beneficial to expand on this section and provide a more comprehensive analysis of the limitations and potential challenges associated with the proposed approach.

**Questions:**

- In Section 3, the authors observe the training imbalance in GCL and propose to address it using POT. Although they present node compactness results, they do not show the InfoNCE loss in the experiments, unlike in Figure 3. It would be helpful to include the InfoNCE loss in the experiment to provide a comprehensive evaluation. Additionally, it would be interesting to investigate if POT can also improve the InfoNCE loss.

- The paper discusses various graph augmentations, and it would be valuable for the authors to provide a more specific discussion on how the proposed POT method can enhance existing GCL methods with different graph augmentations. Exploring the applicability of POT to different augmentation strategies would contribute to a more thorough understanding of its potential benefits.

---

> ### Author Rebuttal · Authors · 2023-08-09
>
> We sincerely thank the reviewer for the positive comments and valuable feedback on our paper. To further address your concerns, we provide additional experiments as well as a more detailed discussion of the limitations.
>
> 1. > In Section 3, the authors observe the training imbalance in GCL and propose to address it using POT. Although they present node compactness results, they do not show the InfoNCE loss in the experiments, unlike in Figure 3. It would be helpful to include the InfoNCE loss in the experiment to provide a comprehensive evaluation. Additionally, it would be interesting to investigate if POT can also improve the InfoNCE loss.
>
>     **Response**: Thanks for pointing this out. In Section 3, we investigate the problem of existing GCL training with InfoNCE, since it is the only metric we have. After that, we propose "node compactness" as a more appropriate and intrinsic metric to evaluate the training of a node in GCL. Therefore, we conduct experiments in the Figures to show the validity of node compactness as well as some properties. That is the reason why we did not provide further analysis of InfoNCE.
>     However, it is still inspiring to investigate whether POT can improve InfoNCE. Intuitively, POT can improve the InfoNCE loss. POT has a larger regularization on the nodes that are not well-trained across all possible augmentations, as a result, the training of nodes under two specific augmentations is also improved. To illustrate this, we conduct an experiment comparing the InfoNCE loss values of the nodes with/without using POT, similar to the experiment in Section 3. We choose GRACE and GCA as baselines and show the result of Cora in Figure 1 of the rebuttal PDF. It can be seen that InfoNCE is improved by POT in two aspects: the average InfoNCE of the nodes is reduced, and the distribution of InfoNCE values becomes more centralized and balanced. Those results and discussions will be added to the Appendix in the revision.
>
> 2. > The paper discusses various graph augmentations, and it would be valuable for the authors to provide a more specific discussion on how the proposed POT method can enhance existing GCL methods with different graph augmentations. Exploring the applicability of POT to different augmentation strategies would contribute to a more thorough understanding of its potential benefits.
>
>     **Response**: Thank you for suggesting improving the experiment. To investigate whether our POT applies to various graph augmentations, we evaluate the performance of POT under four types of topology augmentation: random edge dropping proposed in GRACE; node centrality-based topology augmentations proposed in GCA, including degree centrality, eigenvector centrality, and PageRank centrality. The backbone of the GCL baseline is GRACE. The result on Cora and Flickr is given in Table 1 in the rebuttal PDF. POT consistently outperforms the baseline models with different graph augmentations.
>     As stated in the Limitation section, since more theoretical verifications are needed, POT of other augmentations, including feature augmentations, are working in progress. We are looking forward to more capability of POT.
>
> 3. > The discussion on the limitations of the proposed method is relatively brief. It would be beneficial to expand on this section and provide a more comprehensive analysis of the limitations and potential challenges associated with the proposed approach.
>
>     **Response**: Thanks for your suggestion. We have expanded the limitation section accordingly. Please refer to the "global" rebuttal.

---

### Author Rebuttal · Authors · 2023-08-09

We sincerely thank all reviewers for the acknowledgment of our paper and for many constructive comments. Since some reviewers mentioned that the discussion of limitations may be relatively brief, we expand that part as follows. Due to the limited time, we have tried our best to explore different aspects of possible limitations. A more detailed discussion will be added in the revision.

1. Applying POT to graph-level GCL

   If the downstream task requires graph embeddings, POT still stands. First, related concepts like "graph compactness" can be defined similarly to node compactness. Second, the pooling operators like "MEAN", "SUM", and "MAX" are linear, therefore the form of "graph compactness" can be derived with some modifications on the steps in Appendix A.2.

2. The limitation on the network structure of the encoder

   Definition 5 and Theorem 2 are based on the assumption that the encoder is GCN since GCN is the most common choice in GCL. However, other encoders are possible to be equipped with POT. GraphSAGE[1] has a similar form to GCN, then the derivation steps are similar. For Graph Isomorphism Network[2] and its form is
   $$
   h_v^{(k+1)}=\text{MLP}((1+\epsilon)h_v^{(k)}+\sum_{u\in \mathcal N(v)}h_u^{(k)}).
   $$
   We can first relax the nonlinear activation functions in MLP with Definition 5, then obtain the pre-activation bounds with Theorem 1, and finally follow the steps in Appendix A.2 to derive the form of node compactness when a GIN encoder is applied.

3. The provable training of feature augmentation

   In this paper, we focus on topology augmentations. However, as we have mentioned in the limitation, feature augmentation is also a common class of augmentations. We explore the provable training for feature augmentation as follows. Unlike topology augmentation, the augmented feature is only the input of the first layer, so it is easier to deal with. Inspired by [3], we can relax the discrete manipulation into $l$-1 constraints, then the node compactness is related to the feasible solution of the dual form. A similar binary cross-entropy loss can be designed. More detailed derivations are still working in progress.

However, more efforts are still to be made to deal with some subtle parts.

We greatly thank you again for the reviewers' precious time in reading our paper and rebuttal. We hope that the rebuttal phase can be informative and pleasant for all reviewers.

References:

[1] Hamilton, Will, Zhitao Ying, and Jure Leskovec. "Inductive representation learning on large graphs." Advances in neural information processing systems 30 (2017).

[2] Xu, Keyulu, et al. "How powerful are graph neural networks?." arXiv preprint arXiv:1810.00826 (2018).

[3] Zügner, Daniel, and Stephan Günnemann. "Certifiable robustness and robust training for graph convolutional networks." Proceedings of the 25th ACM SIGKDD International Conference on Knowledge Discovery & Data Mining. 2019.

---

### Decision · Program_Chairs · 2023-09-21

**Decision:**

Accept (spotlight)

**Comment:**

The paper studies the node properties given different graph augmentations in graph contrastive learning. The paper presents a solid idea and nice presentation. All the reviewers agree to accept this paper.